# High-efficiency broadband achromatic metalens for near-IR biological imaging window

Yujie Wang[1,5], Qinmiao Chen[1,5], Wenhong Yang[1,5], Ziheng Ji[1,5], Limin Jin [1], Xing Ma [1], Qinghai Song [1], Alexandra Boltasseva [2], Jiecai Han[3], Vladimir M. Shalaev [2] & Shumin Xiao [1,3,4 ✉]

Over the past years, broadband achromatic metalenses have been intensively studied due to their great potential for applications in consumer and industry products. Even though significant progress has been made, the efficiency of technologically relevant silicon metalenses is limited by the intrinsic material loss above the bandgap. In turn, the recently proposed achromatic metalens utilizing transparent, high-index materials such as titanium dioxide has been restricted by the small thickness and showed relatively low focusing efficiency at longer wavelengths. Consequently, metalens-based optical imaging in the biological transparency window has so far been severely limited. Herein, we experimentally demonstrate a polarization-insensitive, broadband titanium dioxide achromatic metalens for applications in the near-infrared biological imaging. A large-scale fabrication technology has been developed to produce titanium dioxide nanopillars with record-high aspect ratios featuring pillar heights of 1.5 μm and ~90° vertical sidewalls. The demonstrated metalens exhibits dramatically increased group delay range, and the spectral range of achromatism is substantially extended to the wavelength range of 650–1000 nm with an average efficiency of 77.1%–88.5% and a numerical aperture of 0.24–0.1. This research paves a solid step towards practical applications of flat photonics.

[1] State Key Laboratory on Tunable laser Technology, Ministry of Industry and Information Technology Key Lab of Micro-Nano Optoelectronic Information System, Harbin Institute of Technology (Shenzhen), Shenzhen 518055, P. R. China. [2] School of Electrical and Computer Engineering and Birck Nanotechnology Center, Purdue University, West Lafayette 47907 IN, USA. [3] National Key Laboratory of Science and Technology on Advanced Composites in Special Environments, Harbin Institute of Technology, Harbin 150080, P. R. China. [4] Collaborative Innovation Center of Extreme Optics, Shanxi University, Taiyuan 030006 Shanxi, P. R. China. [5] These authors contributed equally: Yujie Wang, Qinmiao Chen, Wenhong Yang, Ziheng Ji. ✉email: shumin.xiao@hit.edu.cn

Miniaturized untethered robots have been intensively studied due to their potential to bring disruptive advances to medical diagnostics and biological studies. Even though significant progress has been made in miniaturizing mobile untethered robots, their size for active imaging ranges from about 1 to 10 mm. The main restriction for the microrobot size is in fact their on-board optical lenses. Consequently, the development of integrated optical lens has been a crucial step towards compact microrobots with on-board functionalities as well as other biomedical imaging techniques such as nano-optical endoscopy for high-resolution optical coherence tomography[1–3]. All-dielectric metalens that is a two-dimensional metamaterial consisting of a large number of dielectric nano-antennas[4–6] offers intrinsic advantages for the realization of miniaturized integrated lens. Metasurface-based devices uniquely focus incident light to a diffraction-limited spot utilizing thin, flatform structures by precisely tailoring the wavefront[7–12]. High focusing efficiency and large numerical apertures (NA) have been demonstrated for a single wavelength soon after the invention of metalens[13–16]. Very recently, the chromatic aberration of metalenses have been successfully tackled too, making them attractive for active imaging[17–23]. The replacement of conventional bulk lenses with all-dielectric achromatic metalenses could address the long-standing challenge in on-board biomedical imaging.

In order to enable practical applications, several metasurface' characteristics must be carefully designed and optimized. To gain ultimate control over the phase of the propagating light, both the metalens' spatial and frequency-dependent phase profile must be controlled. The phase is given by[5]

$$\varphi\left(r, \omega\right) = -\frac{\omega}{c}\left(\sqrt{r^2 + F^2} - F\right) \qquad (1)$$

where $r$, $\omega$, and $F$ are the radial coordinates, angular frequency, and the focal length. To eliminate the chromatic aberration, both the phase $\varphi(r, \omega_d)$ and the group delay $\frac{\partial \varphi}{\partial \omega}|_{\omega = \omega_d}$ around the central frequency $\omega_d$ must be properly controlled in the Taylor expansion of (1). The first term can be controlled by properly designing the resonant response, the Pancharatnam–Berry (PB) phase, or the propagation phase[24–27]. The latter can be manipulated with optimizing the waveguide mode by choosing suitable effective refractive index $n_{eff}$ and thickness $h$[26–28]. Silicon that offers both high refractive index and mature nanofabrication technology, was shown to enable achromatic metalenses with efficiencies up to 50–60% in the spectral range from 1000 to 1800 nm (squares in Fig. 1)[19]. At shorter wavelengths above the band edge, the metasurfaces' performance was, however, strongly limited by the material absorption. Titanium dioxide (TiO₂) is another promising, technologically relevant and CMOS

compatible material that can potentially overcome the absorption limitation of silicon[17,18,29]. However, current TiO₂ metasurface fabrication techniques rely on the atomic layer deposition (ALD) method, which is relatively slow, and restricts the thickness and aspect ratio to the values below 600 nm and 15, respectively[15,17]. Lalanne et al.[30] pioneered an etching technique for TiO₂ nanostructures with a height up to 990 nm and the first perspective for imaging[31]. The aspect ratio is still limited to 8.8–10. As a result, the transmittance of nanopillar is usually sacrificed to fulfill the required group delay. For wavelengths above 630 nm, achromatic metalens' focusing efficiency reduces rapidly to below 30% (dots in Fig. 1). While Ndao et al.[32] reported an average efficiency of ~70%, they sacrificed the focusing ability of the demonstrated metalens and reduced the NA to 0.066. By combining recursive ray-tracing and simulated phase libraries, the hybrid achromatic metalens also shows its potential in achieving high focusing efficiency and broadband achromatism[20]. But the multi-photon process is not ready for shorter wavelength and the photoresist device faces the challenge from long-term durability. Therefore, metalens-based optical imaging in the biological transparency window (650–1000 nm) has so far been severely limited. Moreover, the efficiencies of the demonstrated achromatic metalenses are far below their monochromatic counterparts and cannot convincingly outperform the compound diffractive lens[33–36]. In this work, to address the above-mentioned challenges in metalens realization, we demonstrate high-efficiency, polarization-insensitive, and broadband achromatic metalens operating in the first biological imaging window.

## Results

**Working principle.** In our work, we control the metalens characteristics by carefully designing the group delay, since the material's optical absorption is intrinsic and cannot be modified. The group delay range can be increased through simple geometry change, i.e., by making structures with larger aspect ratio/height. Here, we increase the pillar height to $h = 1500$ nm, which is 2.5 times larger compared to the previous reports. Similar to ref. [20], four types of nanopillars with circular-, ring-, square- and bipolar concentric ring-shaped cross-sections were employed as the metasurfaces building blocks (see Insets in Fig. 2a) to eliminate the polarization dependence. A parameter sweep of the nanopillar size and shape was done to build a library with the values of phase, group delay, and transmittance at the designed operational wavelength of 760 nm (see Supplementary Fig. 2). For an achromatic metalens with the diameter $D = 30$ μm and NA = 0.24, a particle swam optimization method was applied to optimize the nanostructures, minimizing the required group delay

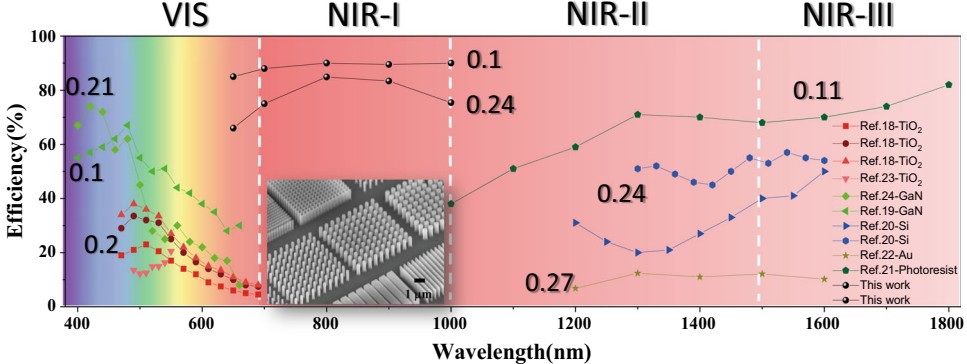

**Fig. 1 Efficiency chart of the demonstrated broadband achromatic metalenses.** The efficiencies of broadband achromatic metalenses with numerical apertures (NA) above 0.1 are plotted in different colors. The results of this work are also presented (black dots). The inset is the tilt SEM image of our TiO₂ nanostructures with pillar height of 1500 nm.

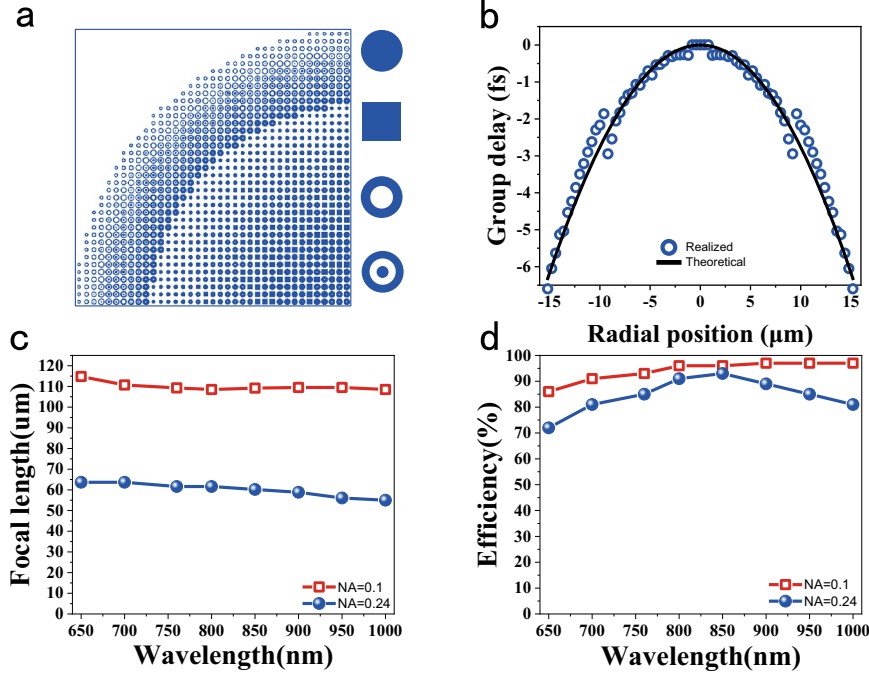

**Fig. 2 Design of the near-IR achromatic metalens. a** The layout of the quarter of the designed metalens. Four fundamental building blocks (unit cells) are enlarged in the insets. **b** The required group delay for broadband achromatism (solid line) and the values provided by the TiO$_2$ nanostructures. **c** and **d** are the numerically calculated focal lengths and efficiencies of TiO$_2$ metalenses with NA = 0.24 (dots) and NA = 0.1 (open squares) at different wavelengths. The diameters of two metalens are 30 and 25 μm, respectively.

range and maximizing the smallest feature size[37]. Figure 2a shows the layout of a quarter of the developed metalens. It consists of four sets of TiO$_2$ nanopillars with different cross-sections and a minimal feature size of 40 nm. The required group delay (black curves) of the achromatic metalens was matched to the propagating group delay in nanopillars (dots in Fig. 2b). As a result, broadband achromatic performance was expected. Figure 2c shows the numerically calculated focal lengths. The incident light with the wavelength 650 to 1000 nm, respectively, is focused by the TiO$_2$ metalens to the designed position with a <7% variation, covering the first biological imaging window. The corresponding focusing efficiency is plotted as dots in Fig. 2d. The minimal and maximal focusing efficiencies are 72% at 650 nm and 93% at 850 nm, respectively. The averaged value in the entire operating spectrum is 83%. Note that the average efficiency can be up to 93% for the small NA of 0.1 and diameter of 25 μm (squares in Fig. 2d and details can be seen in Supplementary Fig. 3), surpassing all the current achromatic metalens demonstrations. In addition, as the nanopillars have at least fourfold rotational symmetry, the achromatic metalens is expected to be polarization insensitive, which is critical for biological applications[38]. Therefore, the developed TiO$_2$ metalens with $h$ = 1500 nm can potentially fill the so-called low-efficiency gap in available achromatic metalens designs.

**Experiments**. Experimental realization of TiO$_2$ metalenses is a challenging task since the state of art pillar height is only 990 nm[6,30]. To overcome this challenge, we developed a top-down etching-based technology for fabricating TiO$_2$ metalenses. First, 1500-nm-thick TiO$_2$ membrane was deposited by the electron beam evaporation and coated with PMMA A2 resist, which was patterned with electron beam lithography. A lift-off process was applied to transfer the nanopatterns to Cr mask. Then, the membrane was etched with the reactive ion etching process (see details in Fig. 3a and "Methods"). After removing the

Cr mask, the TiO$_2$ nanostructures were created. Figure 3b, c depicts the top-view scanning electron microscope (SEM) images of the developed TiO$_2$ metalens. It consists of 4725 nanopillars with four types of different cross-sections. Note that the nanopillars locations, shapes and feature sizes closely follow the numerical design. The tilt-view SEM images of TiO$_2$ metalenses in Fig. 3d, the inset in Fig. 1, and the Supplementary Fig. 5 show that the nanopillars have nearly perfect vertical sidewalls with the measured tilt angle of sidewalls of around 89°−90°. Considering the smallest feature in the metalens, the achieved aspect ratio is around 37.5 or higher (Supplementary Fig. 5). The developed procedure for obtaining high-quality TiO$_2$ nanostructures enabled the experimental realization of the metalens with the designed phase, group delay, and the corresponding high-efficiency broadband achromatism.

The optical properties of the demonstrated TiO$_2$ achromatic metalens were characterized using optical setup in Supplementary Fig. 6. The focal lengths at different wavelengths were obtained by measuring the cross-sectional intensity profiles along the propagation direction (Fig. 3e). In the broad wavelength range from 650 to 1000 nm, the incident light was focused to a bright spot ~60 μm away from the metalens with a small variation. The achieved values of both the focal length and the NA = 0.24 match the numerical design very well.

To demonstrate the realization of a broadband achromatic metalens for the first biological imaging window, Fig. 3f shows the light intensity profiles in the x-y plane at different wavelengths. The focal spot is a circular point at each wavelength. By fitting the intensity distributions along the diameter, all of the calculated Strehl ratios are larger than 0.81 and the full widths at half maximum (FWHM) deviate <9% from the theoretical values, clearly demonstrating that the focal spots are diffraction limited. The 1951 United State Air Force (USAF) resolution test chart was used as a target to test the imaging capability of the developed metalens. For the incident light with the wavelength of 650 nm, the metalens is able to resolve element-4 of group 8, giving a

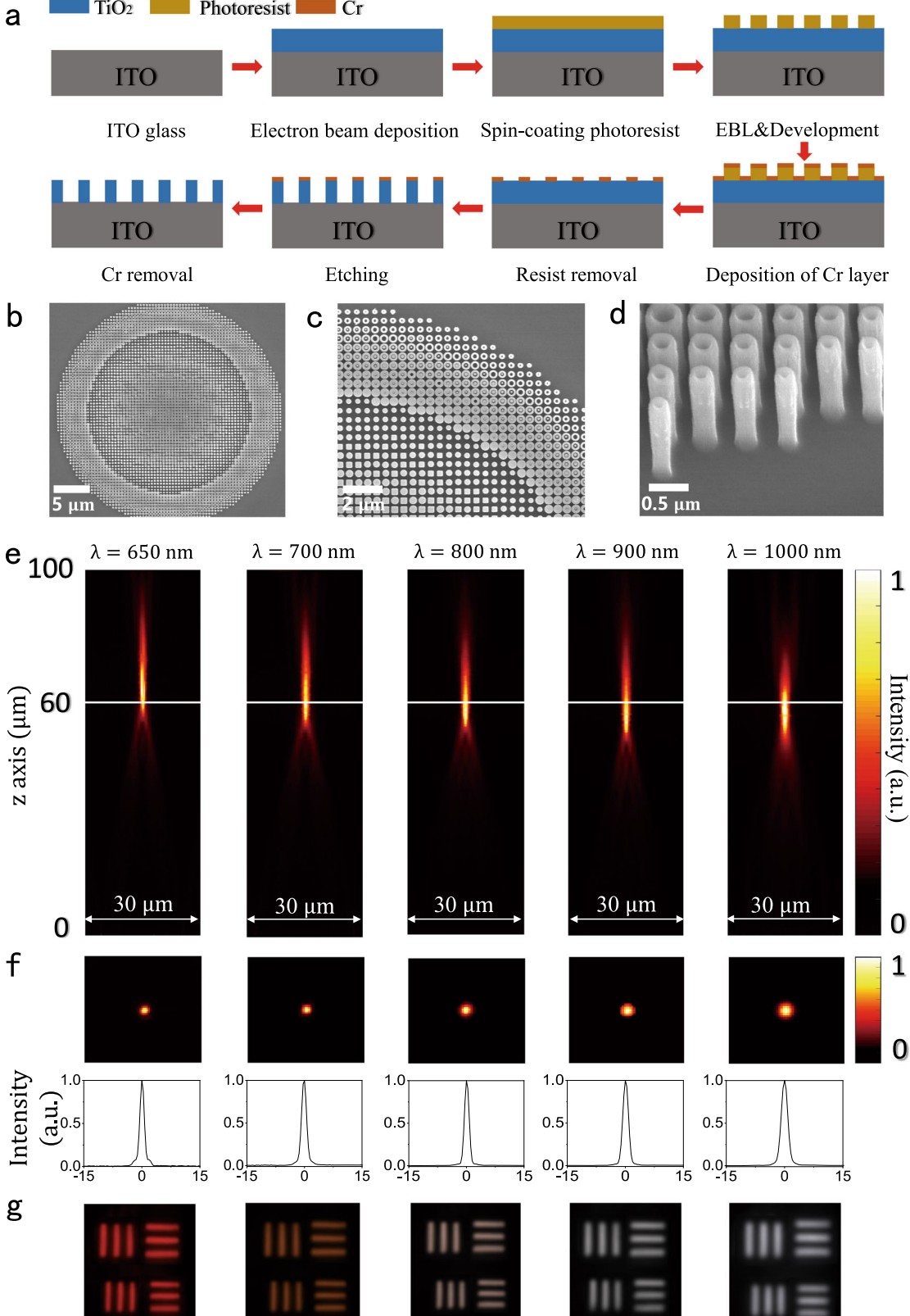

**Fig. 3 The experimentally fabricated TiO₂ metalens with NA = 0.24. a** The schematic of the fabrication process. A highly directional etching process was employed to fabricate TiO₂ nanostructures. **b** and **c** are the top-view SEM images of achromatic metalens with different resolutions. Four types of nanostructures can be clearly identified. **d** The corresponding tilt-view SEM image of the metalens. **e** and **f** are the intensity profiles of focal spots in x-z plane and x-y planes at different wavelengths. The bottom panels depict the intensity distributions along the diameter (solid lines) and the fitted curves (dashed lines). **g** The images of element-6, group-7 of the 1951 Unites States Air Force resolution target recorded by the achromatic metalens.

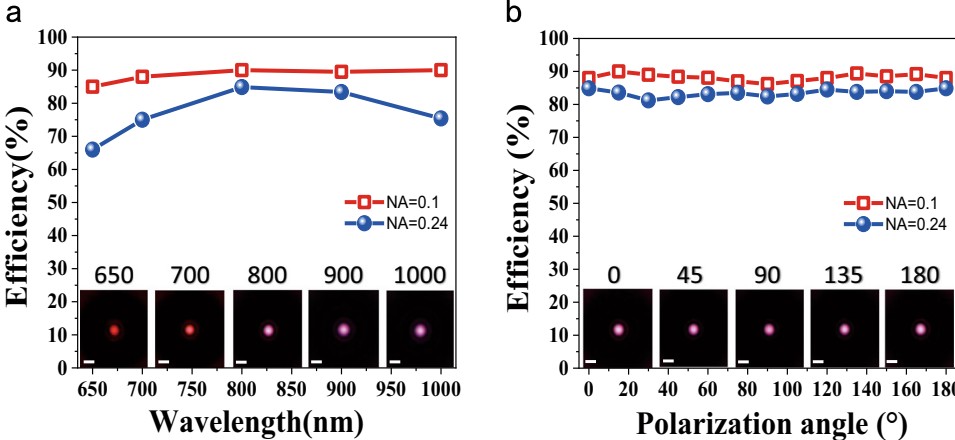

**Fig. 4 Focusing efficiencies of the developed TiO₂ metalenses. a** The experimentally recorded focusing efficiency of the TiO₂ metalens with NA = 0.24 (dots) and NA = 0.1 (open squares) as the function of the incident wavelength. Here, the incident light is un-polarized. **b** The dependence of the focusing efficiencies at 800 nm on the polarization. The insets in (**a**) and (**b**) are images of the focal spots of the metalens with NA = 0.1 at different wavelengths and different polarization, respectively.

resolution above 1.38 µm. While the resolution limit is dependent on the wavelength, the image of element-6 of group 7 was clearly recorded at all wavelengths without tuning the metalens and the target (see Fig. 3g), confirming the broadband achromatism of our metalens very well.

Compared with the achromatism, the focusing efficiency is a more critical criterion for filling the near-IR gap. Similar to refs. [2,8,9], we measured the efficiency by directly comparing the optical power of the focused light with the power of the incident light. The dots in Fig. 4a shows the experimentally recorded efficiencies of the TiO₂ metalens with NA = 0.24. The lowest efficiency is ~65% at 650 nm, whereas the highest value is even ~85% at 800 nm. The averaged efficiency for the entire spectral range is ~77.1%, only a few percent lower than the numerical design. This is consistent with the high-quality fabrication of nanopillars since the efficiency reduces rapidly if the sidewall deviates from 90° in the vertical direction.

Compared with the previous reports, the observed averaged efficiency is record-high. More interesting, we find that the averaged focusing efficiency of the developed TiO₂ metalens can be furthered improved at smaller NAs. When the NA is 0.1, the minimal and maximal focus efficiencies of the metalens at 650 nm and 700 nm are 85% and 90.2%, respectively (see open squares in Fig. 4a and details in Supplementary Fig. 3). The averaged efficiency from 650 to 1000 nm is as high as 88.5%. Since the Fresnel loss at the bottom and etched interfaces are not fully corrected, the averaged high-efficiency can reach above 90% that is superior to the commercial micro-lenses. According to the prediction in refs. [4,5], such high-efficiency broadband achromatic TiO₂ metalens can become a game changer in practical applications of flat photonics.

Polarization insensitivity is another important characteristic for applications in the near-IR window. Figure 4b shows the focusing efficiencies of the demonstrated TiO₂ metalenses at 800 nm at different polarization states. With the change of polarization, the efficiency of metalenses with NA = 0.24 and NA = 0.1 are almost flat at 85 and 89% for different linear polarization and circular polarization, consistent with the results of the non-polarized case in Fig. 4a. Similar polarization-insensitive characteristics hold true for all other wavelengths (see Supplementary Fig. 8).

After we have demonstrated that the developed metalenses can successfully address the low-efficiency gap, we further explored the potential of TiO₂ metalenses in upconversion imaging. In this

experiment, the lanthanide-doped nanocrystals (NCs), which are widely used in bioimaging and labeling, were spread and aggregated on a glass substrate[39]. An additional layer of polystyrene (PS) spheres was deposited on top of the NC clusters. Under a conventional microscope, the NCs clusters were buried in PS spheres and cannot be identified (see Fig. 5a). Then, the continuous wave (CW) laser at 980 nm was focused by the TiO₂ metalens onto the lanthanide-doped NCs clusters. The upconversion fluorescence centered at 655 nm with a linewidth of 20 nm was collected by the same metalens and recorded by a CCD camera (see Supplementary Fig. 9). By scanning the sample with a three-dimensional translation stage, the image can be reconstructed by combing the collected signals for each excitation point. The results are plotted in Fig. 5b. In contrast to conventional microscope image, the sharp edges of the NCs based microplate can be clearly seen in upconversion imaging. The resolution limit is around 1.46 µm, which is determined by the point spread function and the diffraction limit at the emission wavelength. For a direct comparison, the upconversion microscope image has been captured with a commercial achromatic objective lens with NA = 0.26 (Fig. 5c). Both the metalens and the objective lens can capture the structural information of microplate with two-photon excitation with no obvious difference in resolution and intensity distribution (see Supplementary Figs. 10 and 11).

Potential application of the developed TiO₂ achromatic metalens for biological imaging was also evaluated. For that, the HeLa cells containing lanthanide-doped NCs were prepared (Fig. 5d) and optically excited. Figure 5e depicts the fluorescent image of HeLa cells under two-photon excitation. Compared with the bright field microscope image in Fig. 5d, more detailed internal structures of the cell were captured. The image quality and the resolution were close to the ones recorded by the commercial objective lens with the similar NA (Fig. 5f). Thus, the demonstrated achromatic TiO₂ metalens can perform on par with the commercial products for optical imaging. Considering its compactness, flatform, high-efficiency, polarization insensitivity, and diffraction-limit resolution, the proposed achromatic TiO₂ metalenses could trigger a revolution in on-board biomedical diagnosis.

## Discussion

In summary, we developed a top-down fabrication technique for the realization of superior quality, record-high aspect ratio TiO₂

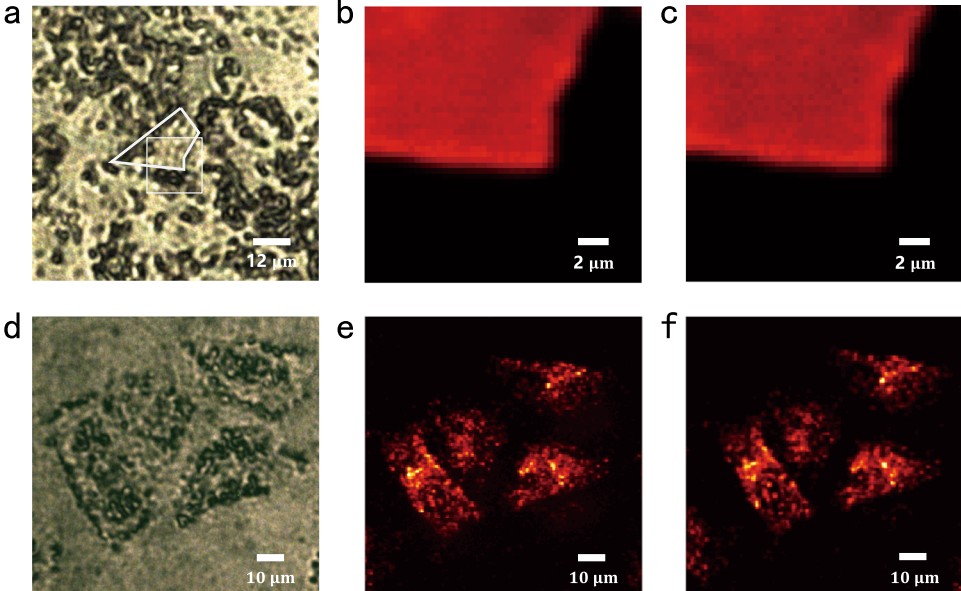

**Fig. 5 Upconversion imaging with the metalens. a** The microscope image of nanocrystals (NCs) on a microplate covered with polystyrene spheres under the white light illumination. The sample is marked with the dashed lines. **b** The upconversion fluorescent image recorded by the achromatic metalens with NA = 0.24. **c** The upconversion fluorescent image recorded by a commercial achromatic objective lens with $N$ = 0.26 (MY10X-823, Mitutoyo). **d** The microscope image of HeLa cells with upconversion NCs. **e** and **f** and the upconversion fluorescent images recorded by the metalens and a commercial objective lens.

metasurfaces. With the increase in the metalens' thickness, the group delay range provided by the $TiO_2$ nanopillars was dramatically enhanced. As a result, high-efficiency, broadband achromatic metalenses were designed and demonstrated in the near-IR biological window for an unpolarized incident light. The focusing efficiencies of the demonstrated achromatic metalenses were as high as 77.1% and 88.5% for the NA of 0.24 and 0.1, respectively, while the maximal focusing efficiency was above 90%. The proposed achromatic metalenses with record-high efficiencies and mass-manufacturable fabrication process could be extended to the visible spectrum as well (see Supplementary Fig. 12) and can potentially trigger a revolution in applications of flat optoelectronics[17,40–43]. Our work is also expected to accelerate the commercialization and biological applications of metasurfaces in portable devices as well as untethered microrobots.

## Methods

**Numerical simulation**. The elements of phase-dispersion library were simulated by a finite element method-based commercial software COMSOL Multiphysics. Each element was simulated under linearly polarized illumination, and the propagation direction of light is along z-axis. The material properties of $TiO_2$ and ITO-coated glass are all considered. Periodic boundary conditions were applied to the x- and y-directions and perfectly matched layer boundary conditions were applied to the z-directions. For each simulation under sweep parameter with respect to the wavelength from 650 to 1000 nm, then the phase spectrum was obtained. We linearly fit the phase spectrum of each element to obtain the group delay. To improve the achromatic performance, the achromatic metalens elements should have good linearity, then any element has an R-squared value <0.995 was dropped. In order to demonstrate the achromatic metalens focusing performance, the metalens was modeled in finite-domain time-difference simulations (Lumerical FDTD Solutions). Perfectly matched layer boundary conditions were used along the transverse and longitudinal directions and under plane wave illumination.

**Deposition of $TiO_2$ film**. The $TiO_2$ film was deposited by electron beam evaporation. The ITO-coated glass substrate is cleaned and placed into the modified electron beam evaporator (Syskey, 30 kV acceleration voltage). The vacuum pressure is pumped to $2 \times 10^{-7}$ Torr and the deposition rate is fixed at 0.6 Å/s. The $TiO_2$ film with 1500 nm thickness is achieved after 6.94 h continuous deposition.

**Fabrication of metalens**. The metalens was fabricated using electron-beam lithography (EBL) on 1500 nm thick titanium dioxide film with quartz substrate to pattern the specific arrays. Firstly, PMMA A2 e-beam resist layer is deposited by spin-coating on titanium dioxide film. Then the sample is exposed through EBL and the patterns are revealed after the development process in MIBK/IPA. Subsequently, 30 nm thick Cr layer as hard mask deposited on sample used electron-gun evaporator and lift-off process is done in solution of PG remover. Next, the patterns are transferred to titanium dioxide film by reactive ion etching (RIE) in Oxford 800 Plus. $SF_6$, $CHF_3$ and $O_2$ are utilized in the etching process. The RF power is 100 W and the pressure is 9 mTorr. The ratio of $CHF_3/SF_6$ is tuned to 5.5, effectively improving the anisotropy of etching. $O_2$ is added to accelerate the etching speed in the vertical direction. The final sample can be obtained after removal of the Cr hard etching mask.

**The optical characterization of metalens**. The metalenses were characterized using a home-built microscope as shown in Supplementary Fig. 6. A collimated laser beam was converted into linearly polarized light after passing through the linear polarizer, Light exiting the polarizer is passing through the objective lens and incident on the metalens, then passes through another objective lens, and finally focuses on the CCD through a tube lens. The longitudinal intensity distribution of the metalens is acquired through a stack of 2D images by changing metalens position in predefined increments. The light intensity distribution of different wavelengths is obtained by inserting color filters.

For focusing efficiency measurement of the metalens, we measured the power in the focal spot (the power of transmission light passing through a circular area with radius of 3*FWHM) and divided by the incident light power (the power of incident light passing through a circular area with radius same as metalens). Note that the efficiency can be further improved by fully correct the Fresnel loss the etched interface and the bottom interface.

## Data availability

All the data supporting the findings of this study are available within the article, its Supplementary Information files, or accessed under the link https://pan.baidu.com/s/4lag6i6F.

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

## Acknowledgements

This research was supported by the National Key Research and Development Project (Grant No. 2018YFB2200403), National Natural Science Foundation of China (Grant Nos. 12025402, 11974092, and 11934012), Shenzhen Fundamental research projects (Grant No. JCYJ20180507184613841, JCYJ20180507183532343, JCYJ20200109112805990, and JCYJ20200109113003946), Shenzhen engineering laboratory on organic-inorganic perovskite devices, and the Fundamental Research Funds for the Central Universities(Grant No. HIT. BRET. 2021009). Purdue authors acknowledge support from the Air Force Office of Scientific Research (AFOSR) grant FA9550-18-1-0002 and FA9550-20-1-0124.

## Author contributions

S.X. designed and supervised the experiment. Y.W. and W.Y. fabricated the samples. Q.C. designed the achromatic metalens. Z.J. and Y.W. performed the optical characterization. L.J. and X.M. provided the upconversion nanoparticles and HeLa cells. Q.S., A.B., J.H., V.S. and S.X. discussed the results and prepared the manuscript.

## Competing interests

The authors declare no competing interests.
