## [Peer Review File · Nature Communications]

High-efficiency broadband achromatic metalens for near-IR biological imaging windowREVIEWER COMMENTS

Reviewer #2 (Remarks to the Author):

In this work, the authors propose using high-aspect-ratio TiO₂ nanopillars to enable achromatic metalenses and try to explore their potential in biological imaging. The paper demonstrates the beautiful fabrication of sample, and the experimental evidence supports the conclusions. However, there are several points that must be carefully addressed before the paper is fit for publication. The comments and necessary changes are outlined below:

1. A seminal work on TiO₂ lens, "Design and fabrication of blazed binary diffractive elements with sampling periods smaller than the structural cutoff." JOSA A 16, 1143-1156 (1999), should be cited. Lalanne et al. fabricated TiO₂ nanopillars with the height of 990nm by top-down etching-based technology, and they also revealed the detail of etching conditions in their paper. However, in this manuscript, the authors claimed that they developed a novel top-down fabrication technique for the realization of superior quality, record-high aspect ratio TiO₂ metasurfaces, but they just showed a traditional EBL with an etching process without detail or related references. The authors should make a comparison between the two works, especially the choice of etching gases for RIE.
2. The authors didn't discuss any design strategies of unit cells, such as how to select the shape of unit cell or how to expand the range of the group delay by tuning the parameters of unit cell. Authors should emphasize if their work is similar/different than other designs of achromatic metalenses, not just the replacement of material. For example, in ref. 17, they also used different shapes of unit cells, such as circular, ring, square, and so on.
3. On page 2 of the main text, the authors claimed that "...which is relatively slow and restricts the thickness to the values below 600 nm. [15] As a result, for wavelengths above 630 nm, metalens' focusing efficiency reduces rapidly to below 30% (dots in Fig. 1). "However, the authors didn't show any evidence that the height of structures leads to low efficiency. And, in ref. 15, Chen et al. claim that "Note that the metalens efficiency is lower compared with our previous works, because to cover a larger range of group delay, some low polarization conversion efficiency elements must be chosen." In addition, in ref. 25, the height of TiO₂ unit cell is only 350nm, but the efficiency of their achromatic metalens is about 70%.
4. I believe that adding more description of Figure 1 will be more helpful for the readers to understand this research field. The authors listed many papers in Figure 1, but only ref. 15 and 17 are discussed in the main text. The working wavelength of achromatic metalens in ref. 18 is also within the wavelength region of Figure 1. I suggest the authors add it in Figure 1. In addition, I suggest the authors label the used materials in Figure 1.
5. In supplementary page 2: " In the main text, the achromatic metalenses are designed following the equations 1 and 2." &" The solid line in Fig. S2B is the required group delay calculated according to Eqs.

(1) and (2) in the main text.” However, there is no equation (2) in the main text or supplementary. Is it Taylor expansion of equation (1)?

6. I suggest the authors classify their unit cells and apply different colors on different kinds of unit cells in Fig. S1. The authors can refer to Figure 2 in ref. 17.

7. The absorption of TiO₂ is very low in the entire visible region. However, the authors didn't expand the working wavelength range to short wavelength (under 630nm). The authors should comment on the limit of their design.

8. The diameter of achromatic metalens is limited by the range of the group delay of unit cells. What is the limit of group delay in this work? What is the diameter limit of metalens with NA 0.24?

9. Why the authors use ITO substrate?

10. What photoresist did the authors use? The authors have different descriptions in the main text page 3, Methods, and supplementary materials. ZEP520A or PMMA?

11. Conventional lenses have chromatic dispersion. Are all the lenses and objectives achromatic in the optical measurement setup?

12. What software is used for numerical simulation? What is the refractive index of the TiO₂ in the simulation?

13. Is the same measurement setup used to measure the 1951 USAF resolution test chart in Figure S5?

Reply to Reviewer #1:

We thank the reviewer for the careful review and valuable suggestions. Based on his/her suggestions and comments, we have carefully revised the manuscript and addressed the comments accordingly. With the suggestions of the reviewer, we believe the quality of this research has been significantly improved. The detailed response can be seen below.

Comment-1: The authors well emphasize the technology jump that they implemented to achieve record performance, see the second method Section and fig 3A. However, to my experience, the performance strongly depends on the quality of the TiO₂ film (TiO₂ films are not easy to deposit; they may be porous ...). Adding details on the deposition technique and the quality of the thick film are welcome. For instance, a measurement of the wavelength-dependent refractive index of the 1.5μm-thick is welcome.

Our Response: We appreciate the reviewer for this valuable comment. The reviewer is absolutely correct that the quality of TiO₂ determines the final quality. In our experiment, the TiO₂ film was deposited by electron beam evaporation. The ITO-coated glass substrate is cleaned and placed into the modified electron beam evaporator (Syskey, 30kV acceleration voltage). The vacuum pressure is pumped down to 2×10^{-7} Torr and the deposition rate is fixed at 0.6 Å/s. The TiO₂ film with 1500 nm thickness is achieved after 6.94 hours continuous deposition. The optical parameters of the deposited TiO₂ film were measured by the ellipsometer and plotted in Fig. R1. The real part of refractive index is above 2.17 in the whole visible and near infrared wavelength range, whereas the imagery part is negligibly small.

Figure R1 (Supplementary Fig. 1 in the Supplementary Information): The refractive index and light extinction coefficient of TiO₂ film.

In the revised manuscript, we have added the deposition process in **Method**. “Deposition of TiO₂ film. The TiO₂ film was deposited by electron beam evaporation. The ITO-coated glass substrate is cleaned and placed into the modified electron beam evaporator (Syskey, 30kV acceleration voltage). The vacuum pressure is pumped to 2

$\times 10^{-7}$ Torr and the deposition rate is fixed at 0.6 Å/s. The TiO₂ film with 1500 nm thickness is achieved after 6.94 hours continuous deposition.”

The optical properties of TiO₂ film were also added as **Supplementary Fig. 1** in the **Supplementary Information**.

Comment-2: The literature on metalenses (and other diffractive lenses) characterization is often debated. Perhaps surprisingly, it is not so simple to properly characterize a small component like the metalenses reported here. The supplementary Section 5 (entitled The optical setups) is quite superficial and does not contain enough quantitative information. I encourage the authors to provide detailed information on how the efficiency is measured.

2a) More details on the setup (focal lens, reference of the doublet ...) should be provided in the supplement.

2b) I think I understood from the main text that the efficiency reported in fig 4 are not corrected from Fresnel loss at the bottom interface (nor at the etched interface). This should be explicitly mentioned in the third method section.

2c) To measure efficiency (and NA) the full lens aperture should be illuminated. The author should explain why it is the case.

2d) More details about the USAF resolution test chart measurements should be provided.

2e) The focusing efficiency measurements are performed by measuring the power in the focal spot through a circular area with radius of 3*FWHM. I am not sure to understand how the measurement is performed. Has a physical aperture been inserted? Is a camera used?

Our Response: We appreciate the reviewer for these valuable comments and suggestions. Following the reviewer’s suggestions, we have added the detail information in the Supplementary Information.

a) The optical setup for characterization of metalens is added **Supplementary Fig. 6** in the **Supplementary Information**. The artificial figure has been replaced with conventional two-dimensional figure with detailed information (see **Fig. R2**). The corresponding description has been added in **Para-1, Page-7** of the **Supplementary Information**. “Basically, a supercontinuum laser is expanded to a collimated beam with a diameter of 5 mm to count the deflection loss at the metalens edge and to avoid the overestimation. After the beam passes the substrate, the light in the metalens region is focused to a diffraction limit spot.”

The optical setup for the resolution chart is added as **Supplementary Fig. 7** (see **Fig. R3**) in the **Supplementary Information**. The corresponding information is added in **Para-2, Page-8** of the **Supplementary Information**. “Supplementary Fig. 7 shows the optical setup for the optical measurement of the 1951 USAF resolution test chart. It is quite similar to Supplementary Fig. 6 except several differences. A supercontinuum laser is focused by a metalens to a diffraction limit spot. In case of narrow band characterization, a bandpass filter can also be applied between metalens and the laser source. The transmitted light through

1951 USAF resolution test chart is collected by a commercial objective lens and imaged to a camera by an additional lens.”

The optical setups for the two-photon imaging have been depicted in **Supplementary Fig. 9** (see Figs. R4 and R5) in the **Supplementary Information**. The corresponding detailed description is added in **Para-1, Page-9** of the **Supplementary Information**. “**Supplementary Fig. 9a** shows the schematic picture of the optical setup for upconversion fluorescent imaging with metalens. The 980 nm laser was spatially filtered into Gaussian profile, then focused to the back focal plane of 50X objective lens by a 300 mm lens, forming a ~50 μm collimated excitation beam at the focal plane of 50X OL to overfill metalens aperture. When the system is working in the monitor module, CCD-2 is placed at 1f position after 250 mm doublet lens, thus conjugated with 50X objective lens focal plane. This module is applied to align metalens and 980 excitation beam, control distance between metalens and sample. Then the system is switched to the confocal module (defined by Metalens, 50X objective lens, 250 mm doublet lens, 100 mm lens and CCD-1) with the flip mirror. Here CCD-1 is placed at 1f position after 100 mm lens, thus conjugated with metalens focal plane. The photoluminescence from the upconversion nanocrystals (NCs) are collected by the same metalens and captured by the CCD-1 camera. A virtual confocal pinhole on the camera will be defined with a sized of 1/3 Airy Unit. Only digital counts within virtual pinhole will be recorded. By scanning the three-dimensional translation state, the two-photon imaging of the NCs based microplate can be monitored even though the microplate is buried by the polystyrene spheres. **Supplementary Fig. 9b** shows the optical setup for the conventional confocal system with objective lens. By removing 300 mm lens in **Supplementary Fig. 9a**, a classical confocal microscope is formed. Here the monitor module can also be saved. Basically, 980 nm laser over filled the 10X achromatic objective lens. The upconversion signals are collected by the objective lens and monitored by the CCD.”

Figure R2 (Supplementary Fig. 6 in the revised manuscript): The optical setup for characterization of TiO₂ metalens. Here metalens is placed at the focal plane of 50X achromatic objective lens.

Figure R3 (Supplementary Fig. 7 in the revised manuscript): The optical setup for the measurement of the 1951 USAF resolution test chart using TiO_2 metalens.

Figure R4 (Supplementary Fig. 9a in the revised manuscript): Up conversion imaging optical setup. It is a confocal scanning microscope consist of two imaging sub-modules, a flip mirror is used to switch between them.

Figure R5 (Supplementary Fig. 9b in the revised manuscript): Up conversion imaging light path for 10X objective lens is a variant of metalens version. By removing 300 mm lens, a classical confocal microscope is formed. 980 nm laser over filled the 10X achromatic objective lens.

b) We thank the reviewer for this valuable suggestion. As depicted in **Fig. R2**, the

light passes the substrate first and then the metalens. In this case, the light is a normally parallel beam and the Fresnel loss is quite similar (but not the same) for the case with or without metalens. The reviewer is absolutely correct that the Fresnel loss at the etched interface is not corrected. This shall make the overall efficiency underestimated. Following the reviewer's suggestion, we have added one sentence in the fourth **Method** section. "Note that the efficiency can be further improved by fully correct the Fresnel loss the etched interface and the bottom interface."

- c) We appreciate the reviewer for this valuable comment. When light passes the edge of metalens, the deflection shall reduce the efficiency. If we only count the central part, the efficiency will be overestimated. In our experiment, the incident laser is expanded to a 5 mm beam to avoid the overestimation. The expanded beam will also perform like a plane wave for the metalens. We have added the corresponding information to **Para-1, Page-7** in the **Supplementary Information**. "Basically, a supercontinuum laser is expanded to a collimated beam with a diameter of 5 mm to count the deflection loss at the metalens edge and to avoid the overestimation. After the beam passes the substrate, the light in the metalens region is focused to a diffraction limit spot."
- d) The optical setup for the measurements of USAF resolution rest chart is shown in **Fig. R3**. The corresponding details have been provided in **Para-2, Page-8** of the **Supplementary Information**. "Supplementary Fig. 7 shows the optical setup for the optical measurement of the 1951 USAF resolution test chart. It is quite similar to Supplementary Fig. 6 except several differences. A supercontinuum laser is focused by a metalens to a diffraction limit spot. In case of narrow band characterization, a bandpass filter can also be applied between metalens and the laser source. The transmitted light through 1951 USAF resolution test chart is collected by a commercial objective lens and imaged to a camera by an additional lens."
- e) We appreciate the reviewer for this valuable comment. The reviewer is correct that a camera was used in the optical characterization. We have added the corresponding information to **Para-2, Page-7** of the **Supplementary Information**. "When the focusing efficiency is measured, a CCD camera is applied. The metalens itself and the light field passing through it can be captured by the imaging system including the objective lens, collection lens, and CCD camera. We first image the metalens and mark the covering region. Then the metalens is in-plane shifted 1 mm. In this case, the incident beam can be captured and the overall intensity is achieved by integrating the intensity within the marked region. The metalens is moved back and the tune the camera to the focal plane of metalens. A bright focal spot can be capture by the CCD camera. To accurately capture the intensity of focal spot, we integrate a region slightly larger than the focal point (3 times of its FWHM). In this experiment, the background is carefully subtracted and the intensity is controlled to be below the saturation of CCD camera."

Comment-3: Appropriate referencing to the earlier literature is quite deficient.

3a) The TiO₂ pillar machinery to fabricate polarization-insensitive high-NA metalenses has been first demonstrated in JOSA A 16, 1143 (1999), and a first perspective for imaging has been offered in J. Opt. A Pure Appl. Opt. 4, S119–S124 (2002). These works report a polarization-insensitive metalens for operation with a VCSEL emitting in the window of interest. The pillars are 990 nm tall. They also reports metagratings etched in a TiO₂ film with a 816 nm thickness for operation in the red. Thus, the two sentences “The state of art technique only produces TiO₂ nanopillars with height of 600 nm” (line 27) and “Experimental realization of TiO₂ metalenses is a challenging task since the ALD technique does not allow achieving thickness above 600 nm [31]” (line 100) are incorrect. Previous state of art for the fabrication of TiO₂ metalenses is not found in ref 31 in my opinion, as highlighted by Table 1 in ref 6.

3b) The idea to control the phase and the group delay by combining different pillar structures with progressively varying sizes is central in this work. It has been first promoted in Optics letters 29, 1593 (2004) to achieve broadband efficiency (only – not to maintain the same focal) and demonstrated in Adv. Opt. Mat. 1, 489–493 (2013).

3c) The sentence “High focusing efficiency and large numerical apertures (NA) have been demonstrated for a single wavelength soon after the invention of metalens. [7-14]” (line 43) dates the invention of metalenses in the last decades according to the references. It is incorrect, see ref 6 for instance.

The four above-mentioned references are quite rooted in the principle of operation and performance of the devices reported in this work, at least much more than many references quoted in the present version manuscript. Consistent and accurate cross-referencing is important.

Our Response: We appreciate the reviewer for the very careful review and valuable suggestions. We appreciate the suggestion of the relevant references from the reviewer. These references are indeed essential for this research field. Following the reviewer’s suggestion, we have carefully revised the manuscript as followings.

a) Two references have been added as **Refs. 30, 31** and highlighted in the introduction (**Para-2, Page-2**) of the revised manuscript. “P. Lalanne et al. pioneered an etching technique for TiO₂ nanostructures with a height up to 990 nm [30] and the first perspective for imaging [31].” The statement in **Para-1, Page-2** of the **Supplementary Information** has been changed to “The state of art technique only produces TiO₂ nanopillars with height of 990 nm”. The sentence in **Para-2, Page-4** of the revised manuscript is changed to “Experimental realization of TiO₂ metalenses is a challenging task since the state of art pillar height is only 990 nm [6, 30].”

30. Lalanne, P. Astilean, S., Chavel, P., Cambriil, E., and Launois, H., Design and fabrication of blazed binary diffractive elements with sampling periods smaller than the structural cutoff. *J. Opt. Soc. Am. A* **16**, 1143-1156 (1999).

31. Lee, M. S. L., Lalanne, P., Rodier, J. C., Chavel, P., Cambriil, E., and Chen, Y.,

Imaging with blazed-binary diffractive elements. *J. Opt. A* **4**, s119–s124 (2002).

b) We thank the reviewer for pointing out these two references, which are added as **Refs. 27, 28** in the revised manuscript. The corresponding expression in **Para-2, Page-2** of the main text has been changed to “The first term can be controlled by properly designing the resonant response, the Pancharatnam–Berry (PB) phase, or the propagation phase [25-28]”.

27. Sauvan, C., Lalanne, P., and Lee, M. S. L., Broadband blazing with artificial dielectrics. *Opt. Lett.* **29**, 1593-1595 (2004).

28. Ribot, C. et al., Broadband and efficient diffraction. *Adv. Opt. Mat.* **1**, 489–493 (2013).

c) We appreciate the reviewer for point out the relevant references. For a fair citation, we have cited Kock’s first paper in microwave and two papers in optical region as **Refs. 7-9**.

7. Kock, W. E., Metallic Delay Lenses. *Bell Syst. Tech.* **34**, 321-339 (1948).

8. Stork, W., Streibl, N., Haidner, H., and Kipfer, P., Artificial distributed-index media fabricated by zero-order gratings. *Opt. Lett.* **16**, 1921-1923 (1991).

9. Ishii, S., Kildishev, A. V., Shalaev, V. M., Chen, K.-P., and Drachev, V. P., Metal nanoslit lenses with polarization-selective design. *Opt. Lett.* **36**, 451-453 (2011).

Comment-4: How the particular pillar geometries shown in fig 2a is chosen? Additional comment on their dispersive properties would be welcome.

Our Response: We appreciate the reviewer for this valuable comment. Since our achromatic metalens is designed for the biological imaging window, it must be polarization insensitive to efficiently capture the photoluminescence from dye or upconversion materials. As a result, we chosen the geometries with 4-fold or rotation symmetry.

Meanwhile, there is a trade-off between numerical aperture (NA), maximal radius (R_{max}), and the frequency range ($\Delta\omega$) following the equation

$$R_{max}NA\Delta\omega \leq 2c\Delta\Phi' ,$$

when the $NA \ll 1$. If we want to improve these parameters (NA, R_{max} or $\Delta\omega$), we have to improve the rang of phase dispersion ($\Delta\Phi'$). In this work, the NA, diameter and broadband have been increased to 0.24, 30 μm and 350 nm (650-1000 nm), and the requirement of $\Delta\Phi'$ need to extend. The $\Delta\Phi' = \left(\frac{d\varphi}{d\omega} \Big|_{max} - \frac{d\varphi}{d\omega} \Big|_{min} \right) \Delta\omega$ is closely related to group delay, which is the derivative with respect to angular frequency:

$$\frac{\partial\varphi(r, \omega)}{\partial\omega} = \frac{1}{c}n_{eff}H + \frac{\omega}{c} \frac{\partial n_{eff}}{\partial\omega} H$$

The range of group delay can be extended by increasing the height and effective index. Further improvement of coverage of group delay requires an increase in the effective index when the height of structure is 1500 nm. However, the group delay provided by the same geometry meta-units with different sizes is finite. We chose five different types units to extend the range of group delay based on our fabrication. A parameter sweep of the units size and shape has been done to build a library with the values of phase and group delay. The phase shift covers 0 - 2π and the group delay varies between 0 fs and 7 fs (see Supplementary Fig. 2).

Figure R6. (Supplementary Fig. 2 in the revised manuscript) The phase shift and group delay libraries.

The corresponding information has been added in **Para-2, Page-3** of the revised manuscript. “Similar to Ref. 20, four types of nanopillars with circular-, ring-, square- and bipolar concentric ring-shaped cross-sections were employed as the metasurfaces building blocks (see Insets in Fig. 2a) to eliminate the polarization dependence.”

The detail information has been added to the **Supplementary Note 2** of **Supplementary Information**.

“We utilize the transmission phase along the TiO₂ nanopillars following the equation

$$\varphi(r, \omega) = \frac{\omega}{c} n_{eff} H, \quad (1)$$

where \$H\$ represent height of TiO₂ structure, and \$n_{eff}\$ represent the effective index. The unit placed at a radial coordinate \$r\$ provides the same phase so that different wavelengths are deflected by the same angle. In order to achieve achromatic focusing, the unit needs to satisfy not only the required phase, but also the group delay. The group delay is the derivative with respect to angular frequency:

$$\frac{\partial \varphi(r, \omega)}{\partial \omega} = \frac{1}{c} n_{eff} H + \frac{\omega}{c} \frac{\partial n_{eff}}{\partial \omega} H \quad (2)$$

The group delay can thus be expanded by increasing the height of nanopillar and the range of effective index. When the pillar height is fixed, the group delay is sensitive to the effective index, which is dependent on the geometry and dimensions in period.

Since we are aiming at the biological imaging window, several key issues must be considered, e.g., the material absorption, the polarization dependence, the efficiency and the achromatic spectral range. Consequently, TiO₂ nanopillars with at least 4-fold rotational symmetry are selected to eliminate the polarization dependence, the material absorption, and to effectively collect the signals. We have increased the pillar height to 1500 nm and fixed the cross sections of nanopillars to four fundamental types, i.e. circle, ring, square and bipolar concentric ring. The phase shift and the group delay of nanopillars with different in-plane sizes are calculated with the finite element methods. The refractive index of TiO₂ film was taken from the optical measurement with the ellipsometer. As shown in **Supplementary Fig. 1**, the TiO₂ has relatively large refractive index of $n > 2.17$ for a large spectral range. The imaginary part of refractive index is negligible in the visible and near infrared spectrum. **Supplementary Fig. 2a** shows the numerical results. The group delay We can see that the phase shift covers $0 - 2\pi$. Meanwhile, the group delay varies between 0 fs and 7 fs. Such kind of library is good enough for generating a broadband, polarization insensitive, large NA and high efficiency metalens.

The efficiency is another important parameter for the design of achromatic metalens. When the pillar height is small such as 600 -800 nm, there is only small range of parameters that can generate the required group delays of 5-7 fs. These parameters typically correspond to the resonant dips or low polarization conversion efficiencies. Once such nanopillars are selected, the overall efficiency of the achromatic metalens reduces quickly. When the pillar height is increased to 1500 nm, the parameter space can be dramatically increased. **Supplementary Fig. 2b** shows the transmittance of 2445 TiO₂ nanopillars as a function of the group delay. While more nanopillars with lower efficiencies appears at larger group delay, there are still numerous TiO₂ nanopillars that can support high efficiency and the required group delay simultaneously. As a result, the overall efficiency of the metalens can be maintained at a high value for the entire spectral range of the first biological window for optical imaging.”

Comment-5: Beautiful SEM pictures are found in the SM. They are not as nice in the main text. Why not trying to insert one in the empty space of fig 1?

Our Response: We appreciate the reviewer for this valuable suggestion. This is a great idea since the blank region can be utilized. Following the reviewer’s suggestion, we have moved one SEM image into Fig. 1 of the revised manuscript. The corresponding information of the inset has also been cited in **Para-2, Page-4** of the revised main text. “The tilt-view SEM images of TiO₂ metalenses in **Fig. 3d**, the inset in **Fig. 1**, and the **Supplementary Fig. 5** show that the nanopillars have nearly perfect vertical sidewalls with the measured tilt angle of sidewalls of around 89° - 90° .”

Figure R7 (Figure. 1 in the main text): Efficiency chart of the demonstrated broadband achromatic metalenses. The efficiencies of broadband achromatic metalenses with numerical apertures (NA) above 0.1 are plotted in different colors. The results of this work are also presented (black dots). The inset is the tilt SEM image of our TiO₂ nanostructures with pillar height of 1500 nm.

Comment-6: Line 83: a particle swarm optimization method. This technical technology (without ref) cannot be understood by the broad readership.

Our Response: We appreciate the reviewer for this valuable suggestion. It is true that the particle swarm optimization method is hard to be understood by the broad readership. To avoid the confusion, we have added the corresponding references in **Para-2, Page-3** of the revised manuscript. “For an achromatic metalens with the diameter \$D = 30 \mu\text{m}\$ and \$NA = 0.24\$, a particle swarm optimization method was applied to optimize the nanostructures, minimizing the required group delay range and maximizing the smallest feature size. [37]”

37. Chen, W. T., Zhu, A. Y., Sisler, J., Bharwani, Z., and Capasso, F., A broadband achromatic polarization-insensitive metalens consisting of anisotropic nanostructures. *Nat. Commun.* **10**, 355 (2019).

Reply to Reviewer #2:

We thank the reviewer for the very careful review and valuable suggestions. Based on his/her suggestions, we have carefully revised the manuscript and all the comments have been addressed accordingly. The detail information can be seen below.

Comment-1: A seminal work on TiO₂ lens, "Design and fabrication of blazed binary diffractive elements with sampling periods smaller than the structural cutoff." JOSA A 16, 1143-1156 (1999), should be cited. Lalanne et al. fabricated TiO₂ nanopillars with the height of 990 nm by top-down etching-based technology, and they also revealed the detail of etching conditions in their paper. However, in this manuscript, the authors claimed that they developed a novel top-down fabrication technique for the realization of superior quality, record-high aspect ratio TiO₂ metasurfaces, but they just showed a traditional EBL with an etching process without detail or related references. The authors should make a comparison between the two works, especially the choice of etching gases for RIE

Our Response: We appreciate the reviewer for this very valuable and insightful comment. We also thank the reviewer for the suggestion of the very important reference. Following the suggestion of the reviewer, "Design and fabrication of blazed binary diffractive elements with sampling periods smaller than the structural cutoff." JOSA A 16, 1143-1156 (1999), has been cited in the revised manuscript as **Ref. 30**.

In the seminar paper, Lalanne et al. fabricated TiO₂ nanopillars with the height of 990 nm by top-down etching-based technology. By applying SF₆/CH₄ with a ratio of 1:1, they have successfully fabricated the TiO₂ nanopillars with a maximal aspect ratio of 8.8-10. As a result, polarization-insensitive blazed binary diffractive components for visible-light have been demonstrated experimentally. This is a milestone work in this research field and is good enough for the fabrication of achromatic metalenses in Refs. 18 and 19.

For the case of achromatic metalens with higher efficiencies and larger numerical apertures, as stated in the main text, TiO₂ nanopillars with larger aspect ratios are required to realize the designed group delay. To realize such aspect ratio, we have replaced the CH₄ with CHF₃ and changed the ratio of SF₆/CHF₃ to 1:5.5. The reduction of SF₆ can effectively reduce the horizontal etching and improve the anisotropy. Note the etching speed is also reduced for such a recipe. To solve this problem, O₂ is also added to enhance the F atom generation and remove the re-deposition of residues on bottom surface. As a result, both of the anisotropy and the etching depth can be improved simultaneously. And an aspect ratio of 37.5 has been experimentally demonstrated.

In the revised manuscript, the seminar work of Prof. Lalanne has been highlighted in **Para-2, Page-2** of the main text. "P. Lalanne et al. pioneered an etching technique for TiO₂ nanostructures with a height up to 990 nm [30] and the first perspective for imaging [31].The aspect ratio is still limited to 8.8-10."

The experimental details have also been added in the **Methods**. "...the patterns are

transferred to titanium dioxide film by reactive ion etching (RIE) in Oxford 800 Plus. SF₆, CHF₃ and O₂ are utilized in the etching process. The RF power is 100 W and the pressure is 9 mTorr. The ratio of CHF₃/SF₆ is tuned to 5.5, effectively improving the anisotropy of etching. O₂ is added to accelerate the etching speed in the vertical direction.”

Comment-2: The authors didn't discuss any design strategies of unit cells, such as how to select the shape of unit cell or how to expand the range of the group delay by tuning the parameters of unit cell. Authors should emphasize if their work is similar/different than other designs of achromatic metalenses, not just the replacement of material. For example, in ref. 17, they also used different shapes of unit cells, such as circular, ring, square, and so on.

Our Response: We appreciate the reviewer for this valuable comment. In our work, we design the achromatic metalens following the design rule in Refs. 15-17. Both of the phase and the group delay are considered. Basically, we utilize the transmission phase along the TiO₂ nanopillars following the equation

$$\varphi(r, \omega) = \frac{\omega}{c} n_{eff} H,$$

where H represent height of TiO₂ structure, and n_{eff} represent the effective index. The unit placed at a radial coordinate r provides the same phase so that different wavelengths are deflected by the same angle. In order to achieve achromatic focusing, the unit needs to satisfy not only the required phase, but also the group delay. The group delay is the derivative with respect to angular frequency:

$$\frac{\partial \varphi(r, \omega)}{\partial \omega} = \frac{1}{c} n_{eff} H + \frac{\omega}{c} \frac{\partial n_{eff}}{\partial \omega} H$$

The group delay can thus be expanded by increasing the height of nanopillar and the range of effective index. When the pillar height is fixed, the group delay is sensitive to the effective index, which is dependent on the geometry and dimensions in period.

Since we are aiming at the biological imaging window, several key issues must be considered, e.g., the material absorption, the polarization dependence, the efficiency and the achromatic spectral range. Consequently, TiO₂ nanopillars with at least 4-fold rotational symmetry are selected to eliminate the polarization dependence, the material absorption, and to effectively collect the signals.

Different from the silicon-based nanostructures in Ref. 20 (Previously labeled as Ref. 17), the fabrication technique for TiO₂ is strongly limited. We have to increase the pillar height to expand the range of group delay. Meanwhile, the smallest feature size is also increased by optimizing the geometry. This will significantly reduce the fabrication difficulty. In our work, the height of TiO₂ nanopillar is fixed at 1500 nm. The dimensions and five geometries have been scanned numerically. Then the TiO₂ nanopillars are selected for the design of achromatic metalens by considering the phase, the range of group delay, and the smallest feature size simultaneously.

In the revised manuscript, we have added one sentence in **Para-2, Page-3** to describe the design of metalens. “Similar to Ref. 20, four types of nanopillars with circular-,

ring-, square- and bipolar concentric ring-shaped cross-sections were employed as the metasurfaces building blocks (see Insets in Fig. 2a)."

In the revised manuscript, we have added the design strategies of unit cells in **Para-3, Page-2** of the **Supplementary Information**. "We utilize the transmission phase along the TiO₂ nanopillars following the equation

$$\varphi(r, \omega) = \frac{\omega}{c} n_{eff} H, \quad (1)$$

where H represent height of TiO₂ structure, and n_{eff} represent the effective index. The unit placed at a radial coordinate r provides the same phase so that different wavelengths are deflected by the same angle. In order to achieve achromatic focusing, the unit needs to satisfy not only the required phase, but also the group delay. The group delay is the derivative with respect to angular frequency:

$$\frac{\partial \varphi(r, \omega)}{\partial \omega} = \frac{1}{c} n_{eff} H + \frac{\omega}{c} \frac{\partial n_{eff}}{\partial \omega} H \quad (2)$$

The group delay can thus be expanded by increasing the height of nanopillar and the range of effective index. When the pillar height is fixed, the group delay is sensitive to the effective index, which is dependent on the geometry and dimensions in period.

Since we are aiming at the biological imaging window, several key issues must be considered, e.g., the material absorption, the polarization dependence, the efficiency and the achromatic spectral range. Consequently, TiO₂ nanopillars with at least 4-fold rotational symmetry are selected to eliminate the polarization dependence, the material absorption, and to effectively collect the signals."

Comment-3: On page 2 of the main text, the authors claimed that "...which is relatively slow and restricts the thickness to the values below 600 nm. [15] As a result, for wavelengths above 630 nm, metalens' focusing efficiency reduces rapidly to below 30% (dots in Fig. 1). "However, the authors didn't show any evidence that the height of structures leads to low efficiency. And, in ref. 15, Chen et al. claim that "Note that the metalens efficiency is lower compared with our previous works, because to cover a larger range of group delay, some low polarization conversion efficiency elements must be chosen." In addition, in ref. 25, the height of TiO₂ unit cell is only 350 nm, but the efficiency of their achromatic metalens is about 70%.

Our Response: We thank the reviewer for the very careful review and the suggestions. The reviewer is absolutely right that we haven't clearly stated the relation between the height and the efficiency. This situation is quite similar to the case of Ref. 18 (Nat. Nanotechnol. 13, 220-226 (2018), Previously numbered as Ref. 15). To simultaneously achieve large NA and large lens diameter in achromatic metalens, a larger range of group delay is therefore required. When the height is fixed, the group delay is sensitive to the effective index, which can be adjusted using the shape and dimensions of the unit cells. Although many structures can provide phase and group delay through controlling the effective index, the transmittance of those structures are usually different a lot. In order to improve focusing efficiency of achromatic metalens, the structures with both high transmittance and accurate phase need to be selected.

In Ref. 18, the height of structures is 600 nm (the same with other previous works). For the nano-fins library they used, most of the high polarization conversion efficiency nano-fins have group delays between 2 to 5 fs. The polarization conversion efficiency for nano-fins with group delays beyond this range will decrease quickly to even zero. As a result, the overall efficiency is decreased rapidly to below 30%.

In our work, we have increased the structure height to 1500 nm, which is 2.5 times larger compared to the previous reports. Due to the increase of height, the effective index is not as sensitive to the change of structure size in the period as Ref. 15. Then a lot of structures with high transmittance can provide group delays between 0 to 7 fs. As shown in **Figure R1**. Most of the nanostructures have transmittance above 90%. The high transmittance of the structures can result in high focusing efficiency of the achromatic metalens.

For the achromatic metalens, the focusing efficiency should be discussed with the other important parameters, e.g., the numerical aperture (NA), the radius of metalens (R_{max}). These parameters are closely related to the range of phase dispersion $\Delta\Phi'$ following the equations

$$R_{max}NA\Delta\omega \leq 2c\Delta\Phi' \quad (1)$$

In Ref.32 (Nat. Commun. 11, 3205 (2020), Previously numbered as Ref. 25), the height of TiO_2 unit cell is only 350 nm. The range of group delay is strongly limited. As a result, both of the NA and the lens radius are restricted too. As depicted in Figs. 4e and 4f in Ref. 32, while the average efficiencies are above 70%, the spot sizes at 1200 nm for three metalens are $\sim 10 \mu\text{m}$, $7 \mu\text{m}$, and $5 \mu\text{m}$, roughly the radius of metalens. In this case, the incident beams are only weakly focused with a NA ~ 0.067 . In our experiment, due to the increase of pillar height, the NA has been increased to 0.24 with a diameter of $30 \mu\text{m}$ and average efficiency of 77.1%. Such kind of NA and device size can be used to image biological objects with acceptable resolution.

Figure R1(Supplementary Fig. 2b in the revised manuscript). The transmittance efficiency versus group delay for different of 2445 elements.

In the revised manuscript, we have changed the statement in **Para-1, Page-3** of the

main text. “the transmittance of nanopillar is usually sacrificed to fulfill the required group delay. For wavelengths above 630 nm, achromatic metalens’ focusing efficiency reduces rapidly to below 30% (dots in Fig. 1).”

The results in **Fig. R1** have also been added as **Supplementary Fig. 2b** and discussed in **Para-2, Page-3** of the **Supplementary Information**. “The efficiency is another important parameter for the design of achromatic metalens. When the pillar height is small such as 600 -800 nm, there is only small range of parameters that can generate the required group delays of 5-7 fs. These parameters typically correspond to the resonant dips or low polarization conversion efficiencies. Once such nanopillars are selected, the overall efficiency of the achromatic metalens reduces quickly. When the pillar height is increased to 1500 nm, the parameter space can be dramatically increased. **Supplementary Fig. 2b** shows the transmittance of 2445 TiO₂ nanopillars as a function of the group delay. While more nanopillars with lower efficiencies appears at larger group delay, there are still numerous TiO₂ nanopillars that can support high efficiency and the required group delay simultaneously. As a result, the overall efficiency of the metalens can be maintained at a high value for the entire spectral range of the first biological window for optical imaging.”

Comment-4: I believe that adding more description of Figure 1 will be more helpful for the readers to understand this research field. The authors listed many papers in Figure 1, but only ref. 15 and 17 are discussed in the main text. The working wavelength of achromatic metalens in ref. 18 is also within the wavelength region of Figure 1. I suggest the authors add it in Figure 1. In addition, I suggest the authors label the used materials in Figure 1.

Our Response: We appreciate the reviewer for this very constructive suggestion. In the revised manuscript, Ref. 21 (Previously numbered as Ref. 18) has been listed in Fig. 1 and the corresponding materials are also labelled as well.

Following the reviewer’s suggestion, we have also discussed more references in **Para-1, Page-3** of the revised manuscript. “By combining recursive ray-tracing and simulated phase libraries, the hybrid achromatic metalens also shows its potential in achieving high focusing efficiency and broadband achromatism. But the multi-photon process is not ready for shorter wavelength and the photoresist device faces the challenge from long-term durability.”

Figure R2. (Figure. 1 in the revised manuscript) Efficiency chart of the demonstrated broadband achromatic metalenses. The efficiencies of broadband achromatic metalenses with numerical apertures (NA) above 0.1 are plotted in different colors. The results of this work are also presented (black dots).

Comment-5: In supplementary page 2: “In the main text, the achromatic metalenses are designed following the equations 1 and 2.” &” The solid line in Fig. S2B is the required group delay calculated according to Eqs. (1) and (2) in the main text.” However, there is no equation (2) in the main text or supplementary. Is it Taylor expansion of equation (1)?

Our Response: We appreciate the reviewer for this very careful review. It is definitely right that the equation (2) is the Taylor expansion of equation (1). During the iteration of the draft, we deleted the equation but forgot to revise the supplementary information accordingly.

We have revised the statement in **Para-3, Page-2** of the **Supplementary Information**. “In the main text, the achromatic metalenses are designed following the equation 1 and its Taylor expansion.”

In **Para-1, Page-4** of the **Supplementary Information**, we have revised the statement as “The solid line in Supplementary Fig. 2b is the required group delay calculated according to Eq. (1) in the main text and its Taylor expansion.”

Comment-6: I suggest the authors classify their unit cells and apply different colors on different kinds of unit cells in Fig. S1. The authors can refer to Figure 2 in Ref. 17.

Our Response: We appreciate the reviewer for this valuable suggestion. Following the suggestion, the phase shift and group delay libraries for different unit cells have been classified using different color as shown in **Figure R3**. The corresponding figure has been replaced in the revised manuscript.

Figure R3 (Supplementary Fig. 2a in the Supplementary Information): The phase shift and group delay libraries. The contributions of different unit cells are labelled with different colors.

Comment-7: The absorption of TiO₂ is very low in the entire visible region. However, the authors didn't expand the working wavelength range to short wavelength (under 630nm). The authors should comment on the limit of their design

Our Response: We appreciate the reviewer for this valuable comment. The absorption of TiO₂ is very low through the whole visible wavelength range even down to the wavelength of 400 nm. From the material absorption, the TiO₂ metalens definitely has the ability of extending its performances to visible spectrum. In our experiment, the achromatic metalens is specifically designed for biological fluorescent imaging and the wavelength range just covers the range through 650-1000 nm, which is already larger than the first near-IR biological imaging window (NIR-I, 750-1000 nm).

The limitation of achromatic metalens is also very clear. According to Ref. 17, the phase gradient of the metalens can be given by:

$$\left| \frac{d\phi}{dr} \right| = \frac{\omega}{c} \frac{r}{\sqrt{f^2 + r^2}} \quad (2)$$

where ω is angle frequency, f is focal length, c is speed of light in vacuum, r is coordinate. In air and at $r = R$, Eq. 2 is simplified to Eq. 3 at the shortest wavelength

$$\left| \frac{d\phi}{dr} \right|_{r=R} = \frac{\omega_{max}}{c} NA = k_{max} NA \quad (3)$$

With a periodicity of P , the phase sample of a 2π phase range is N . Consequently, the phase gradient is limited to:

$$\left| \frac{d\phi}{dr} \right|_{r=R} < \frac{2\pi}{NP} \quad (4)$$

The combination of equations (3) and (4) gives the constraint of

$$NA < \frac{\lambda_{min}}{NP} \quad (5)$$

Equation (5) describes the tradeoff between the NA and bandwidth of achromatic

metalens. The lower bound wavelength is represented as λ_{min} , while the upper bound is implicitly represented in P . If P is too small compared to a desired λ_{max} , the phase-dispersion space is hard to fill. Thus, if we expand the working bandwidth, then it will sacrifice the efficiency and the NA of the achromatic metalens. Meanwhile, the phase sampling N will also be decreased, making the required phase and group delay hard to be fulfilled and thus reducing the efficiency as well. Based on the above description, we have increased the pillar height to 2000 nm and extended the achromatism wavelength range from 400 nm – 1500 nm. However, the average efficiency is $< 30\%$.

If we restrict the operation wavelength from 420 nm to 700 nm, the TiO₂ metalens with pillar height of 1200 nm can have achromatism and high efficiency simultaneously. While keeping the NA and lens diameter at 0.2 and 25 μm , the numerically calculated average efficiency can still be as high as 88.3%. We will present these results with experiments in another work.

In the revised manuscript, we have added the corresponding information in **Para-3, Page-6** of the main text. “The proposed achromatic metalenses with record-high efficiencies and mass-manufacturable fabrication process could be extended to the visible spectrum as well (see Supplementary Fig. 12) and can potentially trigger a revolution in applications of flat optoelectronics.”

The corresponding discussions on the high-performance achromatic metalenses and their restriction are added as a new section of “The working wavelength range” in the **Supplementary Information**.

“As depicted in Fig. S1, TiO₂ has high refractive index and low absorption in the entire visible spectrum the near IR region. In our research, we only demonstrate the one to fill the low efficiency gap in previous achromatic metalens. From the point view of optical properties, TiO₂ has the potential to construct high performance achromatic metalens with a wavelength down to ~400 nm. To demonstrate such a potential, we have also designed a metalens for the visible spectrum. With the reduction of operation wavelength, the thickness of TiO₂ can be reduced to 1200 nm. Based on the same process as the above, we have designed an achromatic metalens with a diameter of 25 \$\mu\text{m}\$ and NA of 0.2. Supplementary Fig. 12a shows a quarter of the metalens. The TiO₂ nanopillars have six types of geometries that have 4-fold or rotational symmetry. The corresponding numerically calculated results are depicted in Supplementary Fig. 12b and Supplementary Fig. 12c. By maintaining the focal length at 60 \$\mu\text{m}\$, the averaged efficiency can be preserved at a higher value of 88.3%.

It is important to note that the spectral range of the achromatic metalens cannot be infinitely extended. In principle, there is a trade-off between the NA, device size, and spectral range for a fixed range of group delay. For all the designs in the manuscript, the group delay is optimized with the high transmittance. As a result, it can maintain the high performance either in the visible spectrum or in the near IR

biological imaging window. There is indeed a possibility of fully covering the visible and near IR spectral range. This can be realized by increasing the range of group delay. Similar to the previous works, we can extend the group delay to a much larger range without considering the transmittance. In this case, we have also designed a metalens for the spectrum from 400 nm to 1500 nm. However, the overall efficiency is strongly spoiled to below 30%. Therefore, we are focusing on the metalens for a particular spectrum in experiments.”

Supplementary Fig. 12: The design of TiO₂ achromatic metalens for the visible spectrum. **a.** The layout of a quarter of the metalens. Six types of nanopillars are employed. **b** and **c** are the numerically calculated focal length and the efficiency. Here the diameter and NA of the metalens are 25 μm and 0.2, respectively.

Comment-8: The diameter of achromatic metalens is limited by the range of the group delay of unit cells. What is the limit of group delay in this work? What is the diameter limit of metalens with NA 0.24?

Our Response: We appreciate the reviewer for this valuable comment. The group delay is the derivative with respect to angular frequency:

$$\frac{\partial \varphi(r, \omega)}{\partial \omega} = \frac{1}{c} n_{eff} H + \frac{\omega}{c} \frac{\partial n_{eff}}{\partial \omega} H$$

The range of group delay can be extended by increasing the height and effective index. When the height is fixed, the group delay is sensitive to the effective index. Effective index can be controlled through adjusting the geometry and dimensions. The height of structure and effective index based on the level of fabrication. In this work, the group delay range covers 0-7 fs by structures with larger aspect ratio/height.

The maximum radius of achromatic metalens is limited by the phase dispersion, and this limitation is:

$$R_{max} \leq \frac{\Delta\Phi'c}{\Delta\omega \left(\frac{1}{NA} - \sqrt{\frac{1}{NA^2} - 1} \right)}$$

when the NA=0.24, the diameter of the achromatic metalens can be up to 30 μm .

In the revised manuscript, the diameter information of two achromatic metalens with NA = 0.24 and NA = 0.1 are added to **the caption of Fig. 2** in the main text. “The diameters of two metalens are 30 \$\mu\text{m}\$ and 25 \$\mu\text{m}\$, respectively.”

Comment-9: Why the authors use ITO substrate?

Our Response: We appreciate the reviewer for this valuable comment. The glass substrate is coated with 13 nm indium tin oxide (ITO) in order to realize conductive layer for electron-beam lithography. The thickness of ITO is chosen as 13 nm in order to decrease the loss. The material properties have been taken into account in the numerical design.

In the revised manuscript, we have added the information in **Method**. “The material properties of \$\text{TiO}_2\$ and ITO-coated glass are all considered.”

Comment-10: What photoresist did the authors use? The authors have different descriptions in the main text page 3, Methods, and supplementary materials. ZEP520A or PMMA?

Our Response: We appreciate the reviewer for this valuable comment and very careful review. PMMA A2 resist has been used as positive photoresist in our experiment. The descriptions in the main text, methods and supplementary information have been unified to PMMA A2 resist.

Comment-11: Conventional lenses have chromatic dispersion. Are all the lenses and objectives achromatic in the optical measurement setup?

Our Response: We appreciate the reviewer for this valuable comment. We used achromatic doublet lens (650~1050 nm) and achromatic objective lens (480~1800 nm) in all multi-wavelength scenario. The only exception is Fig. 3G, where a singlet tube lens was equipped on the camera to image resolution test chart. In principle, the Lens Maker’s Equation for plano-convex lens can gives the focal length on wavelength λ :

$$f_{\lambda} = f_d \cdot \frac{n_d - 1}{n_{\lambda} - 1}$$

where f_{λ} and f_d are the focal length on λ and design wavelength respectively. n_{λ} and n_d are the refractive index on λ and design wavelength respectively. For the BK7 singlet lens we used, $f_d=100$ mm, $n_d=1.5168$, $n_{650}=1.5145$, $n_{1000}=1.5075$. Therefore, f_{650} and f_{1000} are 100.45 mm and 101.83 mm, respectively, giving a focal point shift of 1.38 mm. Meanwhile, the depth of field in image space can be calculated by

$$\text{Axial FWHM}_{\lambda} = 2 \times \frac{\lambda}{NA^2} \times M^2$$

Here NA is the numerical aperture of metalens, M is the magnification of the system

used to image resolution test chart. In our experiment, the magnification M is $M=(100\text{mm}/60\mu\text{m}) / (250\text{mm}/4\text{mm})=26.66$. Consequently, the Axial FWHM_{650nm} and Axial FWHM_{1000nm} are 16.05mm and 24.69 mm. Then we know the introduced focal point shift by the lens is only about 8.6% and 5.6% of axial FWHM for 650 nm and 1000 nm respectively. These values are considered to be negligible in experiment.

Based on the comment of the reviewer, we have purchased the doublet lens to eliminate the chromatic aberration of the tube lens. The experimental results are shown in Fig. R4 below. No obvious improvements in imaging quality can be observed. This is consistent with the above analysis. Since the focal shift of singlet is already more than an order of magnitude smaller than the depth of field, it won't affect the image quality obviously.

To avoid possible confusion, we have **replaced Figure 3g** with the new results in the revised manuscript.

Figure R4. (Figure 3g in the revised manuscript) The images of 1951 USAF resolution target recorded by the achromatic metalens.

Comment-12: What software is used for numerical simulation? What is the refractive index of the TiO₂ in the simulation?

Our Response: We appreciate the reviewer for this valuable comment. The simulations for optical response of unit cell are carried out using COMSOL Multiphysics. The simulation of focusing of metalens is carried out using Lumerical FDTD Solutions. The optical parameters of the TiO₂ are taken from experimental results measured using spectroscopic ellipsometry. As shown in **Figure R5**, the real part of refractive index is above 2.17 in the whole visible wavelength range while the imaginary part is negligible.

Figure R5 (Supplementary Fig. 1 in the revised manuscript). The refractive index and light extinction coefficient of TiO₂ film.

The software information has been added in **Methods** of the revised manuscript. “The elements of phase-dispersion library were simulated by a finite element method based commercial software Comsol Multiphysics.” “...the metalens was modeled in finite-domain time-difference simulations (Lumerical FDTD Solutions)”.

Comment-13: Is the same measurement setup used to measure the 1951 USAF resolution test chart in Figure S5?

Our Response: We appreciate the reviewer for this very careful review. We used a similar setup to measure the 1951 USAF resolution test chart. A supercontinuum laser is focused by a metalens to a diffraction limit spot. In case of narrow band characterization, a bandpass filter can also be applied between metalens and the laser source. The transmitted light through 1951 USAF resolution test chart is collected by a commercial objective lens and imaged to a camera by an additional lens. The detailed schematic of the optical setup is shown in **Fig. R6**.

Figure R6 (Supplementary Fig. 7 in the revised manuscript): The optical setup for the measurement of the 1951 USAF resolution test chart using TiO₂ metalens.

We have added the optical setup as **Supplementary Fig. 7** in the revised supplementary information. The corresponding discussion is also added in **Para-2, Page-8** of the **Supplementary Information**. “Supplementary Fig. 7 shows the optical setup for the optical measurement of the 1951 USAF resolution test chart. It is quite similar to original Fig. S5 except several differences. A supercontinuum laser is focused by a metalens to a diffraction limit spot. In case of narrow band characterization, a bandpass filter can also be applied between metalens and the laser source. The transmitted light through 1951 USAF resolution test chart is collected by a commercial objective lens and imaged to a camera by an additional lens.”

REVIEWER COMMENTS

Reviewer #1 (Remarks to the Author):

The authors have increased the quality of the ms. The later now contains much more precisions on the fabrication and characterization. That's nice.

However, I see one remaining drawback. I find that the introduction does not reflect the state of the art required to understand the originality of the present work. Additionally, I find that it is poorly written:

“All-dielectric metalens that is a two-dimensional metamaterial consisting of dielectric nanoresonators in a Membrane.”

I am not confident with this sentence. I know many metalenses that are not made of nanoresonators, and I know many metasurfaces made of nanoresonators that are not etched in a membrane (if I well understand the word membrane).

“High focusing efficiency and large numerical apertures (NA) have been demonstrated for a single wavelength soon after the invention of metalens. [7-17].”

Ref 7 is performed in the microwaves and Kock is making a high-index metamaterial with metal inclusions and then etch a classical diffractive lens in it. Therefore the link with the present work is somewhat weak. Ref 8 contains a water-wave demonstration. Ref 9 show visible light component with an efficiency of only a few percents. Ref 10 reports an efficiency less than 1%. I am not considering a 1% efficiency a high focusing efficiency... Are all the works in [7-17] reporting large NA?

Minor point: The authors have not understood why I asked to cite refs 27-28 in my previous report.

They say: The first term can be controlled by properly designing the resonant response, the Pancharatnam–Berry (PB) phase, or the propagation phase [25-28].”

In Refs 27-28, not only the effective index n is controlled as in many works, but also the group index $d n / d \omega$ is controlled probably for the first time in a metasurface (this is shown by parameter α in ref 27 that control the variation of n with the frequency in order to make the product $n \omega$ independent of ω at least locally around the operating wavelength). As a result, broadband blazing is achieved. What the present authors do is the same, but in addition they correct the achromatism. Which is for sure more difficult but fundamentally it is the same idea. If the authors disagree, they can simply remove the refs 27-28

I noted one typo in the blue text: scarified

Reviewer #2 (Remarks to the Author):

The authors have addressed all my concerns. I am happy with this version now.

Reply to Reviewer #1:

We thank the reviewer for the careful review and valuable suggestions. Based on his/her suggestions and comments, we have carefully revised the manuscript and addressed the comments accordingly. With the suggestions of the reviewer, we believe the quality of this research has been significantly improved. The detailed response can be seen below.

Comment-1: “All-dielectric metalens that is a two-dimensional metamaterial consisting of dielectric nanoresonators in a Membrane.” I am not confident with this sentence. I know many metalenses that are not made of nanoresonators, and I know many metasurfaces made of nanoresonators that are not etched in a membrane (if I well understand the word membrane).

Our Response: We appreciate the reviewer for the very careful review and valuable suggestions. Following the reviewer’s suggestion, we have carefully revised the manuscript as followings. The corresponding expression in **Para-1, Page-2** of the main text has been changed to “All-dielectric metalens that is a two-dimensional metamaterial consisting of a large number of dielectric nano-antennas”. The expression of nano-antennas is more accurate.

Comment-2: “High focusing efficiency and large numerical apertures (NA) have been demonstrated for a single wavelength soon after the invention of metalens. [7-17].” Ref 7 is performed in the microwaves and Kock is making a high-index metamaterial with metal inclusions and then etch a classical diffractive lens in it. Therefore the link with the present work is somewhat weak. Ref 8 contains a water-wave demonstration. Ref 9 show visible light component with an efficiency of only a few percents. Ref 10 reports an efficiency less than 1%. I am not considering a 1% efficiency a high focusing efficiency... Are all the works in [7-17] reporting large NA?

Our Response: We appreciate the reviewer for the very careful review and valuable suggestions. Following the reviewer’s suggestion, we have separated the references and cited them individually. The corresponding expression in **Para-1, Page-2** of the main text has been changed to “Metasurface-based devices uniquely focus incident light to a diffraction-limited spot utilizing thin, flatform structures by precisely tailoring the wavefront [7-12]. High focusing efficiency and large numerical apertures (NA) have been demonstrated for a single wavelength soon after the invention of metalens. [13-16].” The Ref. 17 has been moved from this position since it is not large NA metalens.

Comment-3: The authors have not understood why I asked to cite refs 27-28 in my previous report. They say: The first term can be controlled by properly designing the resonant response, the Pancharatnam – Berry (PB) phase, or the propagation phase [25-28].” In Refs 27-28, not only the effective index n is controlled as in many works, but also the group index $d n/d\omega$ is controlled probably for the first time in a metasurface (this is shown by parameter α in ref 27 that control the variation of n with the frequency in order to make the product $n\omega$ independent of ω at least locally around the operating wavelength). As a result, broadband blazing is achieved. What the present authors do is the same, but in addition they correct the achromatism.

Which is for sure more difficult but fundamentally it is the same idea. If the authors disagree, they can simply remove the refs 27-28.

Our Response: We appreciate the reviewer for the very careful review and valuable suggestions. We agree with the reviewer and we would like to keep the citation.

Comment-4: I noted one typo in the blue text: scarified

Our Response: We appreciate the reviewer for the very careful review and valuable suggestions. We have corrected this typo. The corresponding expression in **Para-1, Page-3** of the main text has been changed to “the transmittance of nanopillar is usually sacrificed to fulfill the required group delay.”